# Cost Optimal Planning as Satisfiability

## Mohammad Abdulaziz

Techniche Universität München, Germany

### Abstract

We investigate upper bounds on the length of cost optimal plans that are valid for problems with 0-cost actions. We employ these upper bounds as horizons for a SAT encoding of planning with costs. Given an initial feasible plan cost, we experimentally show that this SAT-based approach is able to compute plans with costs better than the initial cost, and in many cases it can match the optimal cost. Also, in multiple instances, the approach is successful in proving that a certain cost is the optimal plan cost.

## Introduction

Compilation to propositional satisfiability (SAT), or other constraint formalisms, has been a successful approach to solving different variants of planning and model checking (Kautz and Selman 1992; Biere et al. 1999). The majority of such compilation based techniques work by submitting multiple queries to a constraints solver, e.g. a SAT solver, and each of those queries encode the question 'Does there exist a witness transition sequence with at most $h$ steps?', where $h$ is some natural number, usually called the horizon. This is repeated for multiple increasing values of $h$. In order for these methods to be complete, there must be an upper bound on $h$, usually called the *completeness threshold*, beyond which no witness could be found if none that is shorter exists. Also, the tighter the bounds, the more efficient these compilation based procedures are.

Previous work has identified different state space topological properties to be completeness thresholds for different variants of model checking and planning problems. E.g. for bounded model checking of safety properties, Biere et al. identified the *diameter*, which is the length of the longest shortest path in the state space, as a completeness threshold. The diameter is also a completeness threshold for SAT-based satisficing planning. Biere et al. also identified the *recurrence diameter*, which is the length of the longest simple path in the state space, as a completeness threshold for bounded model checking of liveness properties. Identifying and computing completeness thresholds was considered an active research area in model checking by Edmund Clarke (Clarke, Emerson, and Sifakis 2009) in his Turing lecture and, indeed, authors have identified completeness thresholds for many involved kinds of model checking prob-lems (Kroening et al. 2011; Bundala, Ouaknine, and Worrell 2012).

Optimal planning is a variant of planning where the solution has to be optimal, according to some measure of optimality. There has been multiple compilations of various types of optimal planning to SAT, satisfiability modulo theories (SMT), and maximum satisfiability formalisms (Büttner and Rintanen 2005; Giunchiglia and Maratea 2007; Robinson et al. 2010; Muise, Beck, and McIlraith 2016; Leofante et al. 2020). Many of the existing compilations tackle optimality criteria of different variants of the length of the plan. Nonetheless, a particularly interesting optimality criterion is plan cost, which has been the primary optimality criterion in planning competitions since 2008 (Gerevini et al. 2009). A gap in the literature seems to be a practical completeness threshold for cost optimal planning problems that have actions with 0-cost. This is one hurdle to the application of SAT-based planning to such problems, since without a reasonable completeness threshold, optimality can only be proved after solving the compilation for a horizon that is the number of states in the state space. This is impractical for most problems since it can be exponentially bigger than the size of the given problem. It should be noted that some approaches try to circumvent the need for a tight completeness threshold, such the ones by Robinson et al. and Leofante et al., which add an over-approximation of the transition relation underlying the planning problem to the encoding. Optimality of a given solution is then proved when this over-approximation is unsatisfiable. Nonetheless, these approaches still need to compute compilations for multiple horizons and they are suscepteble to having to solve compilations for the same exponential horizon, since the over-approximation is generally incomplete, i.e. it could be solvable even if the concrete system is not solvable.

In this work we try to address that gap, and study the suitability of different state space topological properties for being completeness thresholds for cost optimal planning with actions with 0-cost. We identify a completeness threshold that can be practically bounded, and show that no tighter completeness threshold can be computed for a given problem without exploiting cost information, the initial state, or the goal. To test the practical utility of this completeness threshold, we devise a SAT compilation for cost optimal planning, and use that in an any-time planning as satisfiabil-

ity algorithm, where the horizon is fixed from the beginning to the completeness threshold. This algorithm starts with an upper bound on the total cost and improves that cost upper bound every iteration. Experiments show that the algorithm is able to compute plans with costs better than the initial costs, and in many cases it can compute plans whose cost matches the optimal cost. Furthermore, the algorithm is able to prove the optimality of certain costs for a number of instances, some of which could not be proven optimal by the widely used LM-cut planning heuristic.

## Background and Notation

Let $v \mapsto b$ denote a maplet. A mapping $f$ is a finite set of maplets s.t. if $v \mapsto a_1 \in f$ and $v \mapsto a_2 \in f$, we have that $a_1 = a_2$. We write $\mathcal{D}(f)$ to denote $\{v \mid (v \mapsto a) \in f\}$, the domain of $f$. We define $f(v)$ to be $a$ if $v \mapsto a \in f$, and otherwise it is undefined. The composition of two mappings $f$ and $g$, denoted as $f \circ g$, is defined to be $f \circ g = f(g(x))$. In the rest of this paper, we use $|\bullet|$ to denote the cardinality of a set or the length of a list.

We consider planning problems where actions have costs. Such problems are specified in terms of a factored transition system, which is a set of actions, an initial state, a goal, and a cost mapping that assigns costs to actions.

**Definition 1** (States and Actions). *A state, $x$, is a mapping from state-characterising propositions to Booleans, i.e. $\perp$ or $\top$. For states $x_1$ and $x_2$, the union, $x_1 \uplus x_2$, is defined as $\{v \mapsto b \mid v \in \mathcal{D}(x_1) \cup \mathcal{D}(x_2) \wedge$ if $v \in \mathcal{D}(x_1)$ then $b = x_1(v)$ else $b = x_2(v)\}$. Note that the state $x_1$ takes precedence. An action is a pair of states, $(p, e)$, where $p$ represents the* preconditions *and $e$ represents the* effects. *For action $\pi = (p, e)$, the domain of that action is $\mathcal{D}(\pi) \equiv \mathcal{D}(p) \cup \mathcal{D}(e)$.*

**Definition 2** (Execution). *When an action $\pi$ ($= (p, e)$) is executed at state $x$, it produces a successor state $\pi(x)$, formally defined as $\pi(x) =$ if $p \not\subseteq x$ then $x$ else $e \uplus x$. We lift execution to lists of actions $\vec{\pi}$, so $\vec{\pi}(x)$ denotes the state resulting from successively applying each action from $\vec{\pi}$ in turn, starting at $x$.*

We give examples of states and actions using sets of literals, where we denote the maplet $a \mapsto \top$ with the literal $a$ and $a \mapsto \perp$ with the literal $\overline{a}$. For example, $(\{a, \overline{b}\}, \{c\})$ is an action that if executed in a state where $a$ is true and $b$ is false, it sets $c$ to true. $\mathcal{D}((\{a, \overline{b}\}, \{c\})) = \{a, b, c\}$. We also give examples of sequences, which we denote by the square brackets, e.g. $[a, b, c]$.

**Definition 3** (Factored Transition System). *A set of actions $\delta$ constitutes a factored transition system. $\mathcal{D}(\delta)$ denotes the domain of $\delta$, which is the union of the domains of all the actions in $\delta$. Let $\mathsf{set}(\vec{\pi})$ be the set of elements in $\vec{\pi}$. The set of valid action sequences, $\delta^*$, is $\{\vec{\pi} \mid \mathsf{set}(\vec{\pi}) \subseteq \delta\}$. The set of valid states, $\mathbb{U}(\delta)$, is $\{x \mid \mathcal{D}(x) = \mathcal{D}(\delta)\}$.*

**Example 1.** *Consider the factored system $\delta \equiv \{\pi_1 = (\emptyset, \{v_1, v_2\}), \pi_2 = (\emptyset, \{\overline{v_1}, v_2\}), \pi_3 = (\emptyset, \{v_1, \overline{v_2}\}), \pi_4 = (\emptyset, \{\overline{v_1}, \overline{v_2}\})\}$. Figure 1a shows the state space of $\delta_1$, where different states defined on the variables $\mathcal{D}(\delta_1) = \{v_1, v_2\}$ are*

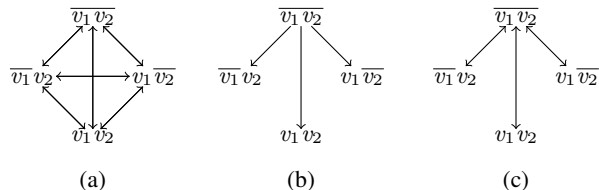

Figure 1: The state spaces of the systems from the different examples.

*shown. Since every state can be reached via one action from every other state, the state space is a clique.*

**Definition 4** (Planning Problem). *A planning problem $\Pi$ is a tuple $(\delta, \mathcal{C}, I, G)$, where $\delta$ is a factored transition system, $I$ is a state s.t. $I \in \mathbb{U}(\delta)$, $\mathcal{G}$ is a state s.t. $\mathcal{G} \subseteq \mathcal{D}(\delta)$, and $\mathcal{C}$ is a mapping from $\delta$ to $\mathbb{N}$. We refer to the different components of $\Pi$ as $\Pi.\delta$, $\Pi.\mathcal{C}$, $\Pi.\mathcal{I}$, and $\Pi.\mathcal{G}$, but when it is unambiguous we only use $\delta$, $\mathcal{C}$, $\mathcal{I}$, and $\mathcal{G}$. A solution to $\Pi$ is an action sequence $\vec{\pi} \in \delta^*$ s.t. $\mathcal{G} \subseteq \vec{\pi}(\mathcal{I})$. We define the function $\mathcal{C}^*$ from $\delta^*$ to $\mathbb{N}$ s.t. $\mathcal{C}^*([]) = 0$, and $\mathcal{C}^*([\pi_1, \pi_2, \ldots]) = \mathcal{C}(\pi_1) + \mathcal{C}^*([\pi_2, \ldots])$. An optimal solution to $\Pi$ is a solution $\vec{\pi}$ s.t. $\mathcal{C}^*(\vec{\pi}) \leq \mathcal{C}^*(\vec{\pi}')$, for any other solution $\vec{\pi}'$ of $\Pi$. For any mapping $f$ from $\mathbb{N}$ to $\mathbb{N}$, we denote by $f(\Pi)$ the planning problem $(\delta, f \circ \mathcal{C}, I, G)$*

**Example 2.** *Let $\mathcal{C} = \{\pi_i \mapsto 1 \mid 1 \leq i \leq 4\}$ be a cost mapping. A planning problem $\Pi$ is $(\delta_1, \mathcal{C}, \{\overline{v_1}, \overline{v_2}\}, \{v_1, v_2\})$. A solution for $\Pi$ is $[\pi_2, \pi_1]$, where $\mathcal{C}^*([\pi_2, \pi_1]) = 2$. An optimal solution for $\Pi$ is $[\pi_1]$, where $\mathcal{C}^*([\pi_1]) = 1$.*

**Definition 5** (Completeness Threshold). *A natural number $CT$ is a completeness threshold for planning problem $\Pi$ iff for any solution $\vec{\pi}$ of $\Pi$ there is a solution $\vec{\pi}'$ s.t. $|\vec{\pi}'| \leq CT$ and $\mathcal{C}^*(\vec{\pi}') \leq \mathcal{C}^*(\vec{\pi})$.*

An evident use for a completeness threshold is in methods for finding cost optimal plans based on compilation to constraints, where one would at most need to unfold the transition relation in the compilation as many times as a completeness threshold for the given problem.

There are multiple possible values that could act as completeness thresholds for a planning problem. The following propositions characterise three such thresholds.

**Proposition 1.** *For any planning problem $\Pi$, $2^{|\mathcal{D}(\delta)|} - 1$ is a completeness threshold for $\Pi$.*

**Proposition 2.** *For any planning problem $\Pi$, if $\vec{\pi}$ is a solution for the problem and if $\mathcal{C}(\pi) = 1$ for every $\pi \in \delta$, then $|\vec{\pi}|$ is a completeness threshold for $\Pi$.*

**Proposition 3.** *For any planning problem $\Pi$, if $\vec{\pi}$ is a solution for the problem and if $\mathcal{C}(\pi) \neq 0$ for every $\pi \in \delta$, then $\lfloor \mathcal{C}^*(\vec{\pi})/c_{\mathsf{min}} \rfloor$ is a completeness threshold for $\Pi$, where $c_{\mathsf{min}}$ denotes $\min_{\pi \in \delta} \mathcal{C}(\pi)$.*

The three above completeness thresholds are either too loose to be of any practical use, or do not hold for planning problems in general. In the next section we study tighter

completeness thresholds for planning problems that can be used with more general planning problems.

## Different Completeness Thresholds

As stated earlier, topological properties of the state space have been employed as completeness thresholds for planning and model checking. In this section we study the suitability of different topological properties as completeness thresholds for planning.

### The diameter

One such topological property is the diameter, suggested by Biere et al. 1999, which is the length of the longest shortest path between any two states in the state space of a system.

**Definition 6** (Diameter). *The diameter, written $d(\delta)$, is the length of the longest shortest action sequence, formally*

$$d(\delta) = \max_{x \in \mathbb{U}(\delta), \vec{\pi} \in \delta^*} \quad \min_{\vec{\pi}(x)=\vec{\pi}'(x), \vec{\pi}' \in \delta^*} |\vec{\pi}'|$$

**Example 3.** *For the transition system $\delta$ from Example 1, the diameter is 1 because any state can be reached from any other state with one action.*

Note that if there is a valid action sequence between any two valid states of $\delta$, then there is a valid action sequence between them, which is not longer than $d(\delta)$. Thus it is a completeness threshold for bounded model-checking and SAT-based planning. There are many features of the diameter that would make it a practically viable completeness threshold. First, it is the tightest topological property of state spaces that has been studied. Secondly, although the worst-case complexity of computing the diameter for a succinct graph is $\Pi_2^P$-hard (Hemaspaandra et al. 2010), there are practical methods that can compositionally compute upper bounds on the diameter (Baumgartner, Kuehlmann, and Abraham 2002; Rintanen and Gretton 2013; Abdulaziz, Gretton, and Norrish 2015, 2017). Unfortunately, the diameter is not a completeness threshold for cost optimal planning, as shown in the following example.

**Example 4.** *Consider the factored system $\delta \equiv \{\pi_1 \equiv (\{\overline{v_1}, \overline{v_2}\}, \{v_1, \overline{v_2}\}), \pi_2 \equiv (\{v_1, \overline{v_2}\}, \{\overline{v_1}, v_2\}), \pi_3 \equiv (\{\overline{v_1}, \overline{v_2}\}, \{\overline{v_1}, v_2\})\}$. Consider the cost mapping $\mathcal{C} \equiv \{\pi_1 \mapsto 1, \pi_2 \mapsto 1, \pi_3 \mapsto 3\}$ where the transitions are labelled with the costs of the corresponding actions. Consider a planning problem $\Pi \equiv (\delta, \mathcal{C}, \{\overline{v_1}, \overline{v_2}\}, \{\overline{v_1}, v_2\})$. The diameter of that system is 1, but there is a plan of length 2 whose cost is less than any plan whose length is bounded by the diameter.*

### The recurrence diameter

Another topological property that has been used as a completeness threshold for different model checking problems is the recurrence diameter, which is the length of the longest transition sequence in the state space of a transition system that does not traverse the same state twice. It was proposed by Biere et al. 1999 as a completeness threshold.

**Definition 7** (Recurrence Diameter). *Let $\mathsf{distinct}(x, \vec{\pi})$ denote that all states traversed by executing $\vec{\pi}$ at $x$ are distinct states. The recurrence diameter is the length of the longest simple path in the state space, formally*

$$rd(\delta) = \max_{x \in \mathbb{U}(\delta), \vec{\pi} \in \delta^*, \mathsf{distinct}(x, \vec{\pi})} |\vec{\pi}|$$

**Example 5.** *For the system $\delta$ from Example 1, the recurrence diameter is 3 as there are many paths with 3 actions in the state space that traverse distinct states, e.g. executing the action sequence $[\pi_1, \pi_2, \pi_3]$ at the state $\{\overline{v_1}, \overline{v_2}\}$ traverses the distinct states $[\{\overline{v_1}, \overline{v_2}\}, \{v_1, v_2\}, \{\overline{v_1}, v_2\}, \{v_1, \overline{v_2}\}]$.*

Note that in general the recurrence diameter is an upper bound on the diameter, and that it can be exponentially larger than the diameter (Biere et al. 1999). However, it can still be exponentially smaller than the number of states in the state space, which would make it a practically useful completeness threshold. The recurrence diameter is a completeness threshold for SAT-based planning and bounded model-checking of safety properties, as well as bounded model-checking of liveness properties, which was the original reason for its inception (Biere et al. 1999).

**Theorem 1.** *For any planning problem $\Pi$, $rd(\delta)$ is a completeness threshold for $\Pi$.*

*Proof.* The proof depends on the following proposition.

**Proposition 4.** *For an action sequence $\vec{\pi} \in \delta^*$, if $\mathsf{distinct}(x, \vec{\pi})$, then $|\vec{\pi}| \leq rd(\delta)$.*

We now show that, given $\vec{\pi} \in \delta^*$ and a state $x \in \mathbb{U}(\delta)$, there is an action sequence $\vec{\pi}'$ s.t. $\mathcal{C}^*(\vec{\pi}') \leq \mathcal{C}^*(\vec{\pi})$, $|\vec{\pi}'| \leq rd(\delta)$, and $\vec{\pi}(x) = \vec{\pi}'(x)$. We do that by complete induction on $\vec{\pi}$. The induction hypotheses would then state that there is such an $\vec{\pi}'$ that can be derived for each $\vec{\pi}_0$, if $|\vec{\pi}_0| < |\vec{\pi}|$. If $\mathsf{distinct}(x, \vec{\pi})$ holds, then the proof is finished by Proposition 4. Otherwise, there are action sequences $\vec{\pi}_1$, $\vec{\pi}_2$, and $\vec{\pi}_3$, s.t. $\vec{\pi}_2$ is not empty, $\vec{\pi} = \vec{\pi}_1 \frown \vec{\pi}_2 \frown \vec{\pi}_3$, and $\vec{\pi}_1(x) = \vec{\pi}_1 \frown \vec{\pi}_2(x)$, where $\frown$ denotes list appending. Since $\vec{\pi}(x) = \vec{\pi}_1 \frown \vec{\pi}_3(x)$, the proof is finished by applying the induction hypothesis to $\vec{\pi}_1 \frown \vec{\pi}_3$. $\square$

A problem with using the recurrence diameter as a completeness threshold is that the complexity of computing it is NP-hard (Pardalos and Migdalas 2004) for explicitly represented digraphs, and that complexity is NEXP-hard (Papadimitriou and Yannakakis 1986) for succinctly represented digraphs, like STRIPS (Fikes and Nilsson 1971). Practically, the existing methods to compute the recurrence diameter have a doubly exponential worst case running time (Kroening and Strichman 2003; Abdulaziz and Berger 2021), and they are only useful when applied to small abstractions in the context of compositionally computing upper bounds on other topological properties. Furthermore, there is not a compositional algorithm that can compute upper bounds on the recurrence diameter using abstractions recurrence diameter. Accordingly, the recurrence diameter cannot

be practically used as a completeness threshold due to the absence of a practical way to compute it or tightly bound it.

## The traversal diameter

Another topological property that was studied in the literature is the *traversal diameter* which was introduced by Abdulaziz 2019. The traversal diameter is one less than the largest number of states that could be traversed by any path.

**Definition 8** (Traversal Diameter). *Let $\mathsf{ss}(x, \overrightarrow{\pi})$ be the set of states traversed by executing $\overrightarrow{\pi}$ at $x$. The traversal diameter is*

$$td(\delta) = \max_{x \in \mathbb{U}(\delta), \overrightarrow{\pi} \in \delta^*} |\mathsf{ss}(x, \overrightarrow{\pi})| - 1.$$

**Example 6.** *Consider a factored system whose state space is shown in Figure 1b. For this system, the traversal diamter and the recurence diameter are both 1. Consider another factored system whose state space is shown in Figure 1c. For this system, the traversal diamter is 3 and the recurence diameter is 2.*

Abdulaziz 2019 showed that the traversal diameter is an upper bound on the recurrence diameter. Since the traversal diameter is an upper bound on the recurrence diameter, it is then a completeness threshold. He also showed that it can be exponentially smaller than the number of states in the state space, and that it can be exponentially larger than the recurrence diameter. Computing the traversal diameter can be done in a worst case running time that is linear in the size of the state space, which is exponentially better than the time needed to compute the recurrence diameter. Furthermore, the traversal diameter is compositionally is via partitioning the set of state variables: an upper bound on the traversal diameter is the product of the traversal diameters of the different projections of the problem's factored transition system on each of the state variables equivalence classes. Although the traversal diameter has the advantage of relatively easy computation with a compositional bounding method, the fact that the traversal diameter is bounded by multiplying all projection traversal diameters leads to computing bounds that are too loose to be of practical value.

## The sublist diameter

Another topological property that can be used as a completeness threshold is the sublist diameter, defined below.

**Definition 9** (Sublist Diameter). *A list $\overrightarrow{\pi}'$ is a sublist of $\overrightarrow{\pi}$, written $\overrightarrow{\pi}' \preceq \overrightarrow{\pi}$, iff all the members of $\overrightarrow{\pi}'$ occur in the same order in $\overrightarrow{\pi}$. The sublist diameter, $\ell(\delta)$, is the length of the longest shortest equivalent sublist to any action sequence $\overrightarrow{\pi} \in \delta^*$ starting at $x \in \mathbb{U}(\delta)$. Formally,*

$$\ell(\delta) = \max_{x \in \mathbb{U}(\delta), \overrightarrow{\pi} \in \delta^*} \min_{\overrightarrow{\pi}(x) = \overrightarrow{\pi}'(x), \overrightarrow{\pi}' \preceq \overrightarrow{\pi}} |\overrightarrow{\pi}'|.$$

**Example 7.** *Consider the factored system $\delta$ from Example 1. For that system the sublist diameter is 1, since for any $\overrightarrow{\pi} \in \delta^*$, executing the last action in $\overrightarrow{\pi}$ will reach the same state reached by executing $\delta$.*

The sublist diameter was first conceived by (Abdulaziz, Gretton, and Norrish 2015) for theoretical purposes, where it was used to show that the diameter can be upper bounded by the projections' topological properties, if the projections were induced by an acyclic variable dependency graph. The way they showed that was by showing that (i) the sublist diameter is an upper bound on the diameter, (ii) the sublist diameter is a lower bound on the recurrence diameter, and, most importantly, (iii) that the sublist diameter can be upper bounded by projections' sublist diameters. Those three properties would make the sublist diameter a very appealing completeness threshold, since it is relatively tight and since it can be upper bounded practically via compositional methods. The following theorem shows that the sublist diameter is indeed a completeness threshold.

**Theorem 2.** *For any planning problem $\Pi$, $\ell(\delta)$ is a completeness threshold for $\Pi$.*

*Proof.* The proof depends on the following proposition.

**Proposition 5.** *For any $\delta$, $x \in \mathbb{U}(\delta)$, and $\overrightarrow{\pi} \in \delta^*$, there is an $\overrightarrow{\pi}'$ s.t. $\overrightarrow{\pi}' \preceq \overrightarrow{\pi}$, $|\overrightarrow{\pi}| \leq \ell(\delta)$, and $\overrightarrow{\pi}(x) = \overrightarrow{\pi}'(x)$.*

Given a solution $\overrightarrow{\pi}$ for $\Pi$, we obtain $\overrightarrow{\pi}'$, which is the witness of Proposition 5. Since $\overrightarrow{\pi}' \preceq \cdot \overrightarrow{\pi}$, we have that the cost of $\mathcal{C}^*(\overrightarrow{\pi}') \leq \mathcal{C}^*(\overrightarrow{\pi})$. This finishes our proof. $\square$

## The subset diameter

As we stated earlier, the sublist diameter has many advantages as completeness threshold, in particular that it is relatively tight and that it has effective methods to compute upper bounds on it. In this section we study how tight can a computed completeness threshold be. Consider the following topological property.

**Definition 10** (Subset Diameter). *A list $\overrightarrow{\pi}'$ is a subset of $\overrightarrow{\pi}$, written $\overrightarrow{\pi}' \subseteq \overrightarrow{\pi}$, iff all the members of $\overrightarrow{\pi}'$ occur in $\overrightarrow{\pi}$. The subset diameter, $\mathcal{S}(\delta)$, is the length of the longest shortest equivalent subset to any action sequence $\overrightarrow{\pi} \in \delta^*$ starting at any state $x \in \mathbb{U}(\delta)$. Formally,*

$$\mathcal{S}(\delta) = \max_{x \in \mathbb{U}(\delta), \overrightarrow{\pi} \in \delta^*} \min_{\overrightarrow{\pi}(x) = \overrightarrow{\pi}'(x), \overrightarrow{\pi}' \subseteq \overrightarrow{\pi}} |\overrightarrow{\pi}'|.$$

**Example 8.** *Consider the factored system $\{\pi_1 \equiv (\emptyset, \{v_1, v_3\}), \pi_2 \equiv (\emptyset, \{\overline{v_1}, v_2\}), \pi_3 \equiv (\emptyset, \{v_1\})\}$. The sublist diameter of this system is 3, because there is not a sublist of the action sequence $[\pi_1, \pi_2, \pi_3]$ that can reach the state $\{v_1, v_2, v_3\}$ from $\{\overline{v_1}, \overline{v_2}, \overline{v_3}\}$. On the other hand, the subset diameter of this system is 2, since $[\pi_2, \pi_1]$ is a subset of $[\pi_1, \pi_2, \pi_3]$ that can reach $\{v_1, v_2, v_3\}$ from $\{\overline{v_1}, \overline{v_2}, \overline{v_3}\}$.*

It should be clear that the following holds.

**Proposition 6.** *For any $\delta$, $d(\delta) \leq \mathcal{S}(\delta) \leq \ell(\delta)$.*

Furthermore, using an argument similar to the one used to prove Theorem 5, we have the following.

**Theorem 3.** *For any planning problem $\Pi$, $\mathcal{S}(\delta)$ is a completeness threshold for $\Pi$.*

More interestingly, we show that the subset diameter is the smallest completeness threshold that can be computed for a planning problem, if the action costs and the initial and goal states are not taken into consideration.

**Theorem 4.** *For any factored transition system $\delta$, there is a planning problem $\Pi$ s.t. $\Pi.\delta = \delta$ and there is a solution $\overrightarrow{\pi}$ for $\Pi$ s.t. $|\overrightarrow{\pi}| = \ell(\delta)$ and for any solution $\overrightarrow{\pi}'$, if $|\overrightarrow{\pi}'| < |\overrightarrow{\pi}|$, then $\mathcal{C}^*(\overrightarrow{\pi}') > \mathcal{C}^*(\overrightarrow{\pi})$.*

*Proof.* Our proof depends on the following proposition.

**Proposition 7.** *For any factored transition system $\delta$, there is a state $x \in \mathbb{U}(\delta)$ and an action sequence $\overrightarrow{\pi} \in \delta^*$ s.t. $|\overrightarrow{\pi}| = \mathcal{S}(\delta)$ and there is not any action sequence $\overrightarrow{\pi}'$ s.t. $\overrightarrow{\pi}' \subseteq \overrightarrow{\pi}$ and $\overrightarrow{\pi}(x) = \overrightarrow{\pi}'(x)$ and $|\overrightarrow{\pi}'| < |\overrightarrow{\pi}|$.*

Obtain a state $x_0$ and an action sequence $\overrightarrow{\pi}_0$ that are the witnesses for Proposition 7. Let $\mathcal{C} = \{\pi \mapsto 0 \mid \pi \in \overrightarrow{\pi}_0\} \cup \{\pi \mapsto 1 \mid \pi \notin \overrightarrow{\pi}_0\}$. We now construct the required planning problem $\Pi$ by letting $x_0$ be its initial state, $\overrightarrow{\pi}(x_0)$ be its goal, $\delta$ be its factored transition system and $\mathcal{C}$ be its cost function. It should be clear that $\overrightarrow{\pi}_0$ is a plan for $\Pi$. Since $x_0$ and $\overrightarrow{\pi}_0$ are the witnesses of Proposition 7, we have that any solution $\overrightarrow{\pi}'$ for $\Pi$ that is shorter than $\overrightarrow{\pi}_0$ will have at least an action not from $\overrightarrow{\pi}_0$. Accordingly, we have that $\mathcal{C}^*(\overrightarrow{\pi}_0) < 1 \leq \mathcal{C}^*(\overrightarrow{\pi}')$, which finishes our proof. $\square$

In this section we primarily focused on the theoretical limit on the tightness of the completeness thresholds and thus devised the subset diameter and showed that it is the tightest. We did not consider on whether the subset diameter can be computed or approximated. Proposition 6 shows that we can use the same methods to bound from Abdulaziz et al. to compute a bound on the subset diameter. However, as shown in Example 8, the subset diameter can be strictly smaller than the sublist diameter, so an interesting open question is whether there is an exponential separation between them, i.e. whether there is a class of factored systems whose subset diameters are exponentially smaller than their sublist diameters. If this were true, an interesting question is whether there are methods to bound or compute the subset diameter that can exploit this tightness.

## A SAT-Encoding for Planning with Costs

To experimentally test the above completeness thresholds, we devise a simple SAT-based encoding of planning with action costs. The core idea of this encoding is to embed action costs into the transition relation by compiling them to their binary representation, effectively keeping track of the plan cost as a part of the state. Previously, more Consider the following compilation of a factored system.

**Definition 11** (Augmented System). *Let, for a natural number $n$, $\mathcal{D}_n$ denote the indexed set of state variable $\{u_1, u_2, \ldots, u_{\lceil \log n \rceil}\}$. Let $x_i^n$ denote the state defined by assigning all the state variables $\mathcal{D}_n$, s.t. their assignments binary encode the natural number $i$, where the index of each*

*variable from $\mathcal{D}_n$ represents its endianess. Note:$x_i^n$ is well defined for $0 \leq i \leq 2^{\lceil \log n \rceil} - 1$. For an action $\pi$, natural numbers $C$, $c$, and $i$, the augmented action $\pi_{i,c}^C$ is defined as $(p \uplus x_i^C, e \uplus x_{i+c}^C)$. For a factored system $\delta$, a natural number $C$, and a function $f$ mapping elements of $\delta$ to natural numbers, the augmented factored system $\delta_f^C$ is defined as $\{\pi_{i,f(\pi)}^C \mid \pi \in \delta \wedge 0 \leq i \leq C - f(\pi)\}$.*

Intuitively, interpreting the function $f$ as a cost function for actions, the augmented factored system is a cost bounded version of the given system, where paths can have at most cost $C$. This is shown in the following example.

**Example 9.** *Consider the factored system and the cost function from Example 4. The augmented system $\delta_\mathcal{C}^2$ would be $\{(\{\overline{v_1, v_2, u_1, u_2}\}, \{v_1, \overline{v_2, u_1}, u_2\}), (\{\overline{v_1, v_2, u_1}, u_2\}, \{v_1, \overline{v_2}, u_1, \overline{u_2}\}), (\{v_1, \overline{v_2, u_1, u_2}\}, \{\overline{v_1}, v_2, \overline{u_1}, u_2\}), (\{v_1, \overline{v_2, u_1}, u_2\}, \{\overline{v_1}, v_2, u_1, \overline{u_2}\})\}$.*

Note that the factored system in the above example will only have paths that, when mapped to the original system, will have a cost of at most 2. Indeed, we have the following theorem which shows how searching for an action sequence whose cost is bounded can be done by searching for any action sequence.

**Theorem 5.** *For a system $\delta$, a mapping $\mathcal{C}$ from $\delta$ to natural numbers, states $x, x' \in \mathbb{U}(\delta)$, and natural numbers $l$ and $i$, there is an action sequence $\overrightarrow{\pi} \in \delta^*$ s.t. $\mathcal{C}^*(\overrightarrow{\pi}) \leq C$, $|\overrightarrow{\pi}| = l$, and $x' \subseteq \overrightarrow{\pi}(x)$ iff there is an action sequence $\overrightarrow{\pi}_C \in \delta_\mathcal{C}^{C*}$ s.t. $|\overrightarrow{\pi}_C| = l$, and $x' \subseteq \overrightarrow{\pi}_C(x \uplus x_i^{C+i})$*

*Proof.* ($\Rightarrow$) We prove this by induction on $\overrightarrow{\pi}$. The base case is trivial. The step case states that $\overrightarrow{\pi} = [\pi] \frown \overrightarrow{\pi}_0$, and the induction hypothesis states that the theorem statement applies to $\overrightarrow{\pi}_0$. Accordingly, we can obtain an action sequence $\overrightarrow{\pi}_C$ by applying the induction hypothesis, after substituting $\pi(x)$ for $x$, $\overrightarrow{\pi}(x)$ for $x'$, $|\overrightarrow{\pi}_0|$ for $l$, $C - \mathcal{C}(\pi)$ for $C$, and $\mathcal{C}(\pi)$ for $i$, s.t. $\overrightarrow{\pi}_C \in \delta_\mathcal{C}^{C*}$, $|\overrightarrow{\pi}_C| = |\overrightarrow{\pi}_0|$, and $\overrightarrow{\pi}(x) \subseteq \overrightarrow{\pi}_C(\pi(x) \uplus x_{\mathcal{C}(\pi)}^C)$. Since $\pi(x) \uplus x_{\mathcal{C}(\pi)}^C = \pi_{0,\mathcal{C}(\pi)}^C(x \uplus x_0^C)$, we have our proof.

($\Leftarrow$) Before we prove this direction, let $x\lfloor_{vs}$ denote the projection of a state on a set of variables $vs$, i.e. $\{v \mapsto b \mid v \in vs \wedge v \mapsto b \in x\}$. Our proof for this direction of the theorem statement is by induction on $\overrightarrow{\pi}_C$. Again, the base case is trivial. The step case states that $\overrightarrow{\pi}_C = [\pi] \frown \overrightarrow{\pi}_0$, and the induction hypothesis states that the theorem statement applies to $\overrightarrow{\pi}_0$. Note that there is an action $\pi' \in \delta$ s.t. $\pi = \pi'^{C+i}_{i,\mathcal{C}(\pi)}$. We can obtain an action sequence $\overrightarrow{\pi}$ by applying the induction hypothesis, after substituting $\pi'(x)$ for $x$, $\overrightarrow{\pi}_C(x)\lfloor_{\mathcal{D}(\delta)}$ for $x'$, $|\overrightarrow{\pi}_0|$ for $l$, $C - \mathcal{C}(\pi')$ for $C$, and $\mathcal{C}(\pi')$ for $i$, s.t. $\overrightarrow{\pi} \in \delta^*$, $|\overrightarrow{\pi}| = |\overrightarrow{\pi}_0|$, $\mathcal{C}^*(\overrightarrow{\pi}) \leq C - \mathcal{C}(\pi')$ and $\overrightarrow{\pi}(x) \subseteq \overrightarrow{\pi}_C(\pi(x) \uplus x_{\mathcal{C}(\pi)}^C)$. Since $\pi'(x) \uplus x_{\mathcal{C}(\pi)}^C = \pi(x \uplus x_0^C)$, we have our proof. $\square$

The theorem above enables solving a bounded cost planning problem with satisficing planning methods.

| Domain | Madagascar Seq | Madagascar ∀ | Madagascar ∃ | No Sym. Seq UNSAT | No Sym. Seq SAT | No Sym. ∀ UNSAT | No Sym. ∀ SAT | No Sym. ∃ UNSAT | No Sym. ∃ SAT | Sym. Seq UNSAT | Sym. Seq SAT | Sym. ∀ UNSAT | Sym. ∀ SAT | Sym. ∃ UNSAT | Sym. ∃ SAT | Total | LM-cut |
|---|---|---|---|---|---|---|---|---|---|---|---|---|---|---|---|---|---|
| **logistics (406)** | 61 29 | 29 13 | 29 13 | 62 29 | 62 29 | 41 41 | 56 33 | 55 37 | 56 37 | 63 29 | 63 29 | 216 41 | 221 35 | 37 33 | 37 30 | 236 41 | 83 46 0 |
| **rover (30)** | 14 4 | 5 4 | 6 4 | 15 4 | 16 4 | 11 4 | 13 4 | 9 4 | 11 4 | 13 4 | 14 4 | 9 4 | 9 4 | 7 4 | 6 4 | 20 4 | 11 10 0 |
| nomystery (24) | 11 10 | 5 2 | 5 2 | 11 10 | 11 10 | 10 10 | 10 10 | 10 10 | 11 10 | 11 9 | 11 9 | 9 10 | 9 10 | 7 6 | 7 6 | 11 10 | 6 3 7 |
| **zeno (50)** | 19 15 | 9 7 | 13 7 | 19 15 | 19 15 | 19 13 | 19 13 | 20 11 | 19 13 | 22 15 | 22 15 | 35 13 | 35 13 | 18 11 | 22 11 | 40 15 | 28 21 0 |
| hiking (20) | 7 5 | 3 1 | 2 1 | 7 5 | 7 5 | 5 5 | 5 4 | 5 5 | 5 4 | 7 5 | 7 5 | 10 — | 10 — | 4 2 | 4 2 | 16 5 | 5 1 4 |
| **Transport (40)** | — | — | — | — | — | — | — | — 1 | — 1 | 9 — | 9 — | 32 — | 32 — | 7 — | 8 — | 33 1 | 8 0 1 |
| **woodworking (20)** | 2 — | 1 — | 2 — | 3 — | 7 — | 4 — | 6 — | 4 — | 5 — | 3 — | 5 — | 2 — | 2 — | 1 — | 1 — | 9 0 | 11 0 0 |
| visitall (50) | 42 17 | 15 8 | 15 8 | 42 17 | 42 17 | 20 14 | 22 14 | 20 14 | 22 14 | 42 17 | 42 17 | 20 15 | 25 15 | 20 15 | 25 15 | 42 17 | 16 16 1 |
| satellite (10) | 6 4 | 6 4 | 6 3 | 8 5 | 10 5 | 9 5 | 10 5 | 9 5 | 10 4 | 9 5 | 10 5 | 9 6 | 9 5 | 7 5 | 7 4 | 10 6 | 10 9 0 |
| scanalyzer (20) | 1 1 | 1 — | 2 1 | 3 1 | 4 1 | 3 1 | 3 1 | 3 1 | 3 1 | 6 1 | 6 1 | 9 1 | 10 1 | 3 1 | 3 1 | 11 1 | 8 3 1 |
| tidybot (47) | 16 10 | 7 1 | 7 1 | 16 10 | 16 10 | 8 1 | 8 1 | 8 1 | 8 1 | 13 9 | 13 9 | 7 7 | 7 7 | 8 7 | 7 7 | 16 10 | 24 13 0 |
| **trucks (2)** | 2 — | — | — | 2 — | 2 — | — | 2 — | 2 — | 2 — | 2 — | 2 — | 2 — | 2 — | — | — | 2 0 | 2 0 0 |
| maintenance (5) | 5 5 | 5 3 | 5 3 | 5 5 | 5 5 | 5 5 | 5 4 | 5 5 | 5 5 | 5 5 | 5 5 | 5 5 | 5 5 | 5 5 | 5 5 | 5 5 | 0 0 5 |
| Parking (40) | — | — | — | — | — | — | — | — | — | 38 — | 39 — | — | — | — | — | 39 0 | 2 0 0 |
| floortile (14) | — | — | — | 1 — | 2 — | — | 1 — | 2 — | 3 — | — | 1 — | 1 — | 1 — | — | — | 3 0 | 8 0 0 |
| barman (22) | 2 1 | — | — | 2 1 | 2 1 | 1 1 | 1 1 | 1 1 | 1 1 | 2 1 | 2 1 | 2 1 | 2 1 | — | — | 3 1 | 1 1 0 |

Table 1: Each column represents a configuration of SAT encoding and SAT solving and shows two numbers: the number of problems for which the cost was improved, and the number of problems for which a certain cost was proved optimal. The first column has the domain name and the number of instances for which Fast Downward was able to compute an upper bound, and the domain name is bold if it has instances with 0-cost actions. Columns 2-4 represent data problems solved with Madagascar's SAT-solver, where each column represents one encoding. The next 12 columns represent data for problems solved using Kissat, with and without symmetry breaking, and for the two configurations of Kissat, SAT and UNSAT. The second to last column represents the number of different problems whose initial cost was improved by all combinations, and those proven to be optimal. The last column shows (i) how many problems were optimally solved by Fast Downward using the LM-cut heuristic, (ii) on how many problems does Algorithm 1 match the optimal cost as computed by LM-cut, and (iii) for how many instances could Algorithm 1 prove a cost is optimal, where LM-cut failed.

**Definition 12** (Augmented Problem). *For a planning problem* Π, *a natural number* C, *the augmented planning problem* $\Pi^C$ *is defined as* $(\delta_\mathcal{C}^C, \mathcal{C}, I \uplus x_0^C, G)$.

**Corollary 1.** *For a system* Π *and a natural number* l, *there is a solution* $\vec{\pi}$ *for* Π *s.t.* $\mathcal{C}^*(\vec{\pi}) \le C$ *and* $|\vec{\pi}| = l$ *iff there is a solution* $\vec{\pi}_C$ *for* $\Pi^C$ *s.t.* $|\vec{\pi}_C| = l$.

## Any-Time SAT-Based Optimal Planning

To find an optimal plan, we need to iteratively decrement the cost bound until no plan is found. A challenge to doing that is that the size of the augmented system is a factor of $C$ larger than the original system, where $C$ is the cost upper bound. One way to circumvent this size increase employs the following proposition.

**Proposition 8.** *For a set of natural numbers* N, *let* gcd(N) *denote their greatest common divisor. For a planning problem* Π *let* gcd(Π) *denote* gcd({$\mathcal{C}(\pi) \mid \pi \in \delta$}). *An action sequence* $\vec{\pi}$ *is a solution for* Π *with cost* C *iff* $\vec{\pi}$ *is a solution for* Π/ gcd(Π) *with cost* $\lfloor C/ \text{gcd}(\Pi) \rfloor$.

Using the above proposition to scale down the action cost bound dramatically limits the blow up in the size of the augmented factored systems for many domains. Another way to limit the size of the augmented factored system is by factoring the actions of the augmented system.

Algorithm 1 is the overall algorithm that we use. It is an any time algorithm that, given an initial plan, computes plans with improving costs until the optimal cost is reached. That algorithm assumes that there is a SAT-based procedure *solve* that computes a satisficing plan, given a planning problem and a horizon. It also assumes that there is a procedure *factor* that factors actions in a planning problem, i.e. if there are two actions $\pi_1$ and $\pi_2$ s.t. $\pi_1 = (\{v\} \cup p, e)$ and $\pi_2 = (\{\overline{v}\} \cup p, e)$ in $\delta$, both of the actions are removed and replaced by $(p, e)$, where this is greedily done until a fixed point is reached. Lastly, it also uses a function to compute the completeness threshold with every iteration since the completeness threshold might change depending on the current plan cost, if the problem has all unit cost actions (Proposition 2), or if it has no 0-cost actions (Proposition 3). That function is specified in the following corollary.

**Corollary 2.** *For a planning problem* Π *and a solution* $\vec{\pi}$ *for* Π, *let* $CT(\vec{\pi}, \Pi)$ *be* $|\vec{\pi}|$ *if* $\mathcal{C}(\pi) = 1$ *for every* π ∈ δ, $\lfloor \mathcal{C}^*(\vec{\pi})/c_{\min} \rfloor$ *if* $\mathcal{C}(\pi) \ne 0$ *for every* π ∈ δ, *and* $\ell(\delta)$ *otherwise. A completeness threshold for* Π *is* $CT(\vec{\pi}, \Pi)$.

## Experimental Evaluation

We experimentally test Algorithm 1 to investigate how capable it is to (i) find plans with better costs than the initial plan, (ii) find plans with optimal costs, and (iii) show that a plan is an optimal plan. We implement the function *solve* by computing the a SAT encoding using the SAT-based planner Madagascar (Rintanen, Heljanko, and Niemelä 2006), where

**Algorithm 1:** Input: plan $\vec{\pi}$ and a problem $\Pi$.

$\vec{\pi}' := \vec{\pi}$
**while** $\vec{\pi}' \neq$ **none**
   $\vec{\pi} := \vec{\pi}'$
   $\Pi' := factor((\Pi/\gcd(\Pi))^{\lfloor \mathcal{C}^*(\vec{\pi})-1/\gcd(\Pi) \rfloor})$
   $\vec{\pi}' := solve(\Pi', CT(\vec{\pi}, \Pi))$
**return** $\vec{\pi}$

we try the three different possible encodings computed by Madagascar: the sequential, the $\forall$-step, and the $\exists$-step encodings. To solve the formulae resulting from these encodings, we use the SAT solver of Madagascar and the state-of-the-art SAT solver Kissat (Biere et al. 2020). Furthermore, we use Kissat's two configurations: SAT and UNSAT, and we run the experiments once with adding symmetry breaking clauses using the tool BreakId (Devriendt et al. 2016) and once without them. The initial plans are computed by the planner Fast Downward (Helmert 2006) with the FF heuristic (Hoffmann and Nebel 2001). To compute completeness threshold when there are action with 0-cost, instead of computing the sublist diameter, we use upper bounds computed using previously published methods (Abdulaziz, Gretton, and Norrish 2017; Abdulaziz 2019; Abdulaziz and Berger 2021). The initial plan computation, completeness threshold computation, and the execution of Algorithm 1 are given 1800s timeout and 4GB memory limit. As a baseline, we use Fast Downward with the LM-cut heuristic (Pommerening and Helmert 2012) to compute optimal plans, with the same time and memory limits.

Table 1 shows the coverage of the different configurations of SAT encoding and SAT solving. It shows that none of the configurations is consistently the best in all domains, whether in terms of proving optimality, or improving the initial cost. Nonetheless, it seems to always that configurations using Kissat as a SAT solver outperform the configurations where Madagascar's SAT solver is used in more domains. Also it seems that the different configurations are complementary to each other within each of the domains, which is why the total number of solved instances is better than the number of instances solved by any individual configuration in 10 domains out of 16.

Another point to note is that, overall, Algorithm 1 proves optimality for less problems than Fast Downward with the LM-cut heuristic. Interestingly, nonetheless, Algorithm 1 is able to prove optimality on instances on which LM-cut fails like in NoMystery, Hiking, Transport, Visitall, Scanalyzer, and Maintenance. We note that all of these domains have no 0-cost actions. Furthermore, Algorithm 1 is able to compute plans with costs that match those computed using LM-cut, but without being able to prove that these are the optimal costs. This is the case in Logistics, Rover, Zeno, Satellite, Scanalyzer, and TidyBot.

To get a more fine-grained view of the quality of computed plans, the plot in Figure 2 shows the cost of the cheap-

est plan computed by all of the configurations and compares it to the cost of the initial plan. In this plot, we have restricted ourselves to problems where the initial bound was at most 100 to preserve readability of the plots. The problems shown in that figure show that the costs are significantly improved for many of the domains.

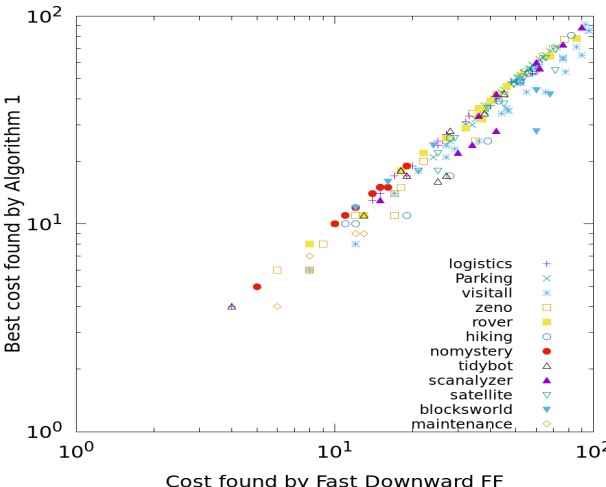

Figure 2: Comparison of costs of initial plans computed by Fast Downward with FF vs Algorithm 1.

## Discussion

In this work we have investigated different completeness thresholds for cost optimal planning. These completeness thresholds enable more applicability of SAT-based planning techniques to cost optimal planning, in particular to problems that have actions with cost 0. We devised a simple SAT-based technique that effectively operates by compiling away action costs into action effects. Experimental results using our method show reasonable performance. Although its coverage is less than state-of-the-art A* based optimal planners, using SAT-based techniques for cost optimal planning problems has multiple advantages. E.g. it is very easy to obtain a certificate of optimality if the SAT solver proves a certain cost is optimal, which is a problem that recently attracted attention (Eriksson, Röger, and Helmert 2017). Also, it can be easily adapted to generating different plans with the same cost, namely, by adding constraints to the SAT-encoding that prohibit a given plan, which is another interesting problem (Katz et al. 2018).

There are multiple interesting future directions in which this work can be further pursued. First the upper bounds could be improved by either incorporating the action costs, initial state, or goal. Another interesting problem is to find whether there is an exponential separation between the subset diameter and sublist diameter and, if there is one, investigating methods to compute or bound the subset diameter. The encoding can also be improved by employing approximate methods when compiling the costs, e.g. the method by Hoffmann et al. 2007 could be adapted to compiling costs. Also, since our experiments show that the different combinations of SAT-encoding and SAT solving are com-

plementary, a portfolio approach can be used to optimise the used combination for different instances.

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
