# OpenReview forum: "Cost Optimal Planning as Satisfiability"
_icaps-conference.org/ICAPS/2021/Workshop/HSDIP — HSDIP 2021_

### Official Review · AnonReviewer1 · 2021-05-26

**Confidence:** 4
**Overall Score:** Accept

**Review:**

The paper investigates SAT-based optimal planning for tasks with 0-cost
actions. SAT-based planning works by iteratively checking whether plans of
incremental length exist, and proving optimality is hard for them since the
first plan found (the one with shortest length) might not be optimal. The paper
presents several existing and one new upper bound for optimal plan length, as
well as a SAT encoding that will find optimal plans when presented with such an
upper bound. An experimental evaluation shows that the approach is able to
improve solution costs and often also able to prove optimality. It is also
compared against LM-cut and while LM-cut overall performs better the SAT-based
techniques can sometimes show optimality where LM-cut fails.

The paper is well written, especially the section about the different upper
bounds is well explained and supported by helpful examples when needed. The
theoretical part is correct as far as I can tell and the evaluation convinces
me that the approach is viable and promising. Overall a great paper, I clearly
recommend to accept it.

I do have some minor questions and comments about the description of the final
algorithm:
1) Does the solve procedure from Algorithm 1 perform only one SAT call or is it
   incremental calls with lengths 1 .. bound?
2) In Corollary 2, why do we use the sublist diameter rather than the subset
   diameter? Is it because the computation of the subset diameter is not
   fleshed out yet? Or is it a typo?
3) Do we really need to call the factor procedure within the loop? To my
   understanding the problem (unit cost/0-cost/gcd) doesn't change during the
   while loop and the diameters should not change either, right?
4) When factoring two actions, don't you need to consider their cost as well?


Minor comments:
 - The traversal diameter: "the traversal diameter is compositionally is
 via..." -> is compositional via...?
 - Definition 9/10: Maybe explicitly define what equivalent an equivalent
 sublist/subset is (I assume it's about reaching the same state when applying
 the path)
 - Theorem 4: Shouldn't it be $|\pi| = \mathscr L(\delta)$?
 - Proposition 7: It is not obvious to me why this proposition is true
 - A SAT-Encoding for Planning with Costs: "Previously, more Consider..." ->
 Consider ...
 - Theorem 5 is a bit hard to read: It is not very clear why we restrict
 the length of the path to l, and why we use C+i for the cost encoding (I don't
 have a better suggestion though).
 - Experimental Evaluation: "it seems to always that..." -> "it seems that
 configurations ... always outperform ..."
 - Figure 2: diagonal line missing
 - References: page numbers missing in almost every entry

---

### Official Review · AnonReviewer2 · 2021-05-27
**.**

**Confidence:** 4
**Overall Score:** Accept

**Review:**

The paper discusses a wide range of methods for finding upper bounds on the length of plans including zero-cost actions, and it proposes a method for applying these upper bounds to SAT-based planning in order to find cost-optimal plans.

Although the paper is theory-heavy, it is a nice read. I especially liked how the examples are used to gain the intuition behind the presented concepts. Also, I find the way action costs are compiled into the SAT representation interesting, and experimental evaluation well executed. Overall, I suggest to accept the paper.

Minor issues:

page 4, first paragraph under "The traversal diameter": "introduced by Abdulaziz 2019" -- missing parenthesis around "2019"

page 4, paragraph under Example 6: "... diameter is compositionally is via ..." -- typo

page 4, right column, first paragraph: "...they showed that was by showing that ..." -- I would suggest to reformulate the sentence

page 5, paragraph under Proposition 7: "... showed it is the tighest." -- You mean tightest of the discussed ones, right?

page 5, first paragraph under "A SAT-Endcoding for Planning with Costs":  "Previously, more Consider the ..." -- typo

Proposition 8: The notation "\Pi / gcd(\Pi)" wasn't introduced, or if it was at this point I already forgot what it means. So I would suggest to at least remind readers what it means.

page 6, right column, last paragraph: "... computing the a SAT..." -- typo

Discussion, last paragraph: Hoffman et al. 2007 -- missing parenthesis around "2007"

Discussion, last paragraph: "compiling costs" --> compiling out costs

---

### Decision · Program_Chairs · 2021-06-10

**Decision:**

Accept

**Comment:**

Congratulations, all reviewers agree that the paper is a clear accept.

Please, address the minor issues pointed out by the reviewers in your camera-ready.